| Open Peer Review | Antimicrobial Chemotherapy | New-Data Letter

# Gene dosage of *PDR16* modulates azole susceptibility in *Candida auris*

Trinh Phan-Canh,[1,2] Tamires Bitencourt,[1,3] Karl Kuchler[1,2]

**KEYWORDS** *Candida auris*, Pdr16, drug resistance, azole resistance

*Candida auris* is a human fungal pathogen of utmost medical relevance exhibiting exceptionally high resistance to all clinically used antifungal drugs, including azoles, polyenes, and echinocandins as well as flucytosine (1–4). The marked skin tropism and pronounced adherence to catheters or other medical devices facilitate outbreaks in hospital settings (5–7). Most importantly, up to 34% and 4% of clinical *C. auris* isolates show multi- or pan-antifungal resistance, respectively, to three classes of antifungal drugs (2, 8, 9). Thus, the WHO declared *C. auris* as one of the top four critical fungal pathogens demanding immediate attention in research and drug discovery (10). Although insights about molecular resistance mechanisms of *C. auris* are increasing, it remains unclear how *C. auris* establishes multidrug resistance (MDR) traits without significant fitness trade-offs (11–14). Of note, some MDR *C. auris* clinical isolates show dysregulated expression of the *PDR16* (B9J08_004982) gene encoding a putative fungal phosphatidylinositol transfer protein (15, 16). Here, we show that *PDR16* modulates susceptibility to different antifungal drug classes including azoles.

Multiple sequence alignments of Pdr16 showed that the primary sequence is highly conserved in fungal species from *Saccharomyces* to *Candida* spp. (Fig. S1A). The pairwise alignment between *C. auris* and *C. albicans* Pdr16 using the emboss_needle alignment tool revealed a 62.9% identity and 73.7% similarity based on sequences from the Candida Genome Database (17). Hence, Pdr16 function may also be conserved in *C. auris*. Therefore, we ablated *PDR16* in the multidrug-resistant recipient strain 462/P/14 (R) by replacing the coding region with a dominant nourseothricin (*NAT1*) marker. Furthermore, we generated a gain-of-function mutant by ectopic overexpression of Pdr16 (e*PDR16*) in the drug-sensitive strain 2431/P/16 (S) using the strong *ENO1* enolase promoter from *C. auris* (16). The mRNA levels of *PDR16* in e*PDR16* were approximately 64-fold higher when compared to the wild-type (WT) strain S (Fig. S1B).

We then performed standard broth dilution antifungal susceptibility assays following the Clinical and Laboratory Standards Institute method (18) for azoles, amphotericin B (AMB), caspofungin (CSF), and flucytosine (5FC). Notably, e*PDR16* increased the minimum inhibitory concentrations (MIC) for all antifungal drugs tested except for CSF (Fig. 1A; Fig. S1C; Table 1). Specifically, e*PDR16* increased MICs for azoles up to 16-fold, while deletion of *PDR16* in strain R reduced MICs by about 4-fold (Fig. 1A; Fig. S1C). Although deletion of *PDR16* in strain R did not significantly change MICs for AMB and 5FC, overexpression increased AMB MICs by 4-fold and 2-fold for 5FC. The MIC for CSF was not affected; however, e*PDR16* cells showed better growth at supra-MIC concentrations compared to the WT S (Fig. 1A, CSF). Interestingly, e*PDR16* levels did not significantly affect mRNA levels of typical resistance transporters, including *CDR1*, *CDR2*, *MDR1*, and *SNQ2* upon voriconazole treatment (Fig. S1B). These results suggest that Pdr16 may be involved in multiple mechanisms contributing to combinatorial antifungal responses, particularly azole susceptibility.

Address correspondence to Karl Kuchler, kuchlerkarl1@gmail.com.

The authors declare no conflict of interest.

See the funding table on p. 4.

10.1128/spectrum.02659-24   **1**

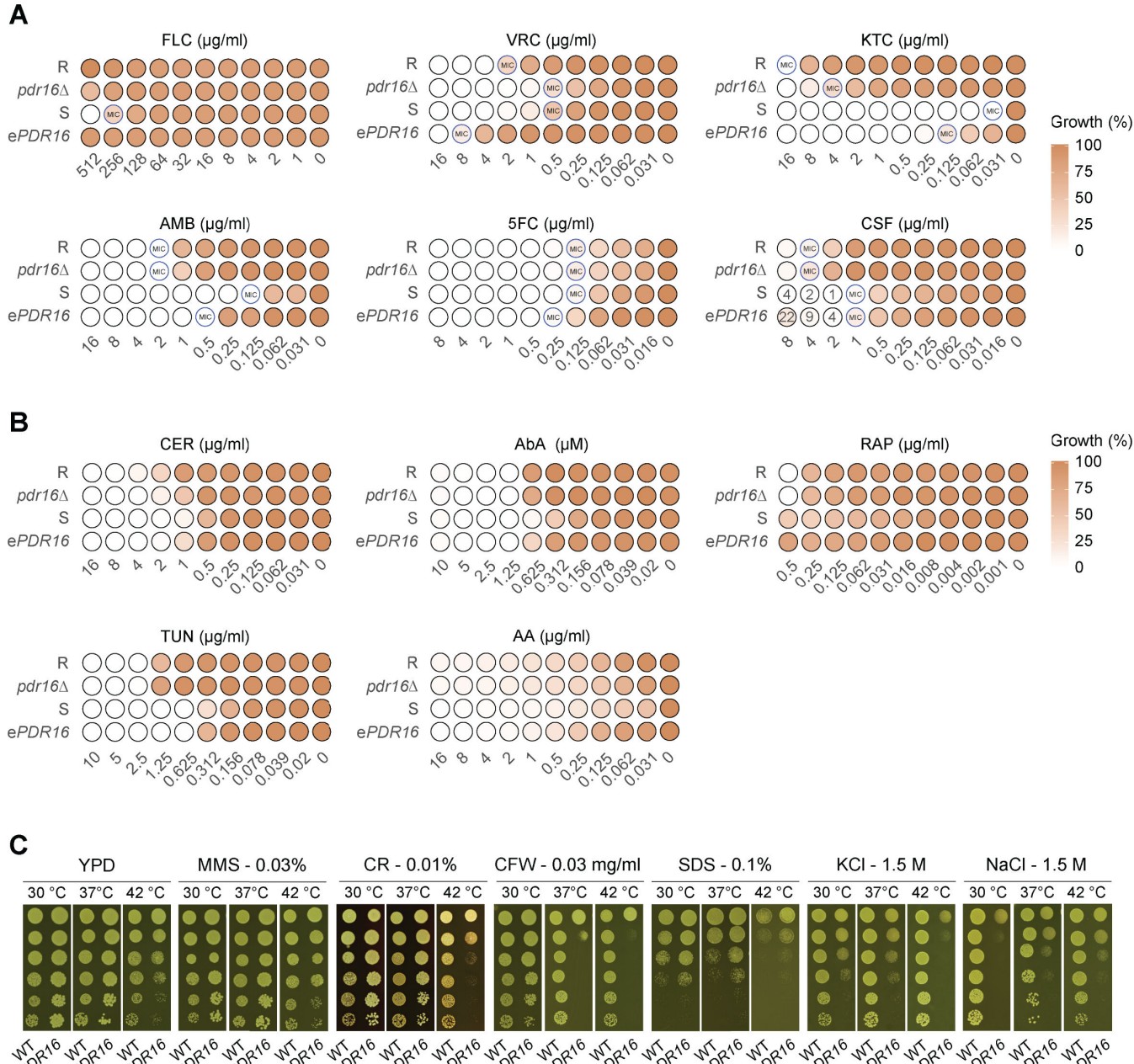

**FIG 1** *PDR16* modulates antifungal drug susceptibility in *C. auris*. (A–B) Dose response minimum inhibitory concentration (MIC) assays of WT and mutant strains for different antifungal drugs and stress agents in RPMI medium buffered at pH seven with MOPS at 37°C. Three different transformants of each mutant were tested. The MIC for each strain, indicated by a blue circle, is defined as the concentration at which 50% growth inhibition is visible when compared to the control, except for AMB, which are wells where MIC indicates 100% inhibition. (C) Agar plate spotting assays at different temperatures and with stress agents. Suspensions with $5 \times 10^7$ *C. auris* cells/mL were 5-fold serially diluted into 96-well plates; aliquots of 2 µl cell suspensions were spotted onto different agar plates; wild-type (WT); ectopic overexpression of *PDR16* (e*PDR16*). Antifungal drugs included fluconazole (FLC), voriconazole (VRC), ketoconazole (KTC), amphotericin B (AMB), caspofungin (CSF), and flucytosine (5FC). Inhibitors targeted endoplasmic reticulum (tunicamycin [TUN]), fatty acid and sphingolipid biogenesis (cerulenin [CER], aureobasidin A [AbA]), mitochondria (antimycin A [AA]), mTOR signaling (rapamycin [RAP]), genotoxic stressor (methyl methane sulfonate [MMS]), cell wall (calcofluor white [CFW], Congo Red [CR]), cell membrane function and osmosensitivity (sodium dodecyl sulfate [SDS], potassium chloride [KCl], sodium chloride [NaCl]). Data are from at least three biological replicates.

To test this hypothesis, we treated the *C. auris* with various inhibitors targeting specific cellular components. Given the narrow effective window for some inhibitors, we also conducted spotting assays instead of broth dilution assays (Fig. 1C; Fig. S1D).

**TABLE 1** Minimal inhibitory concentrations (µg/mL) of *Candida auris* strains to different antifungal drugs[a]

| Strains | FLC | VRC | KTC | AMB | 5FC | CSF |
|---|---|---|---|---|---|---|
| Parental R | >512 | 2 | 16 | 2 | 0.125 | 4 |
| *pdr16Δ* | ~512 | 0.5 | 4 | 2 | 0.125 | 4 |
| Parental S | 256 | 0.5 | 0.031 | 0.125 | 0.125 | 1 |
| *ePDR16* | >512 | 8 | 0.125 | 0.5 | 0.25 | 1 |

[a]Fluconazole (FLC), voriconazole (VRC), ketoconazole (KTC), amphotericin B (AMB), flucytosine (5FC), caspofungin (CSF).

Corresponding to slight effects for CSF susceptibility, e*PDR16* caused hypersensitivity to cell wall stressors such as Congo red (CR, a β-glucan-binding agent) and calcofluor white (CFW, a chitin-binding agent) when compared to the control strain S (Fig. 1C). CSF blocks the Fks1 β-1,3-D-glucan synthase, which is required for synthesizing cell wall β-1,3-D-glucan (19, 20). These data suggest that *PDR16* levels affect cell wall integrity, possibly explaining the altered growth of e*PDR16* under CSF stress at supra-MICs. This observation is consistent with our previous findings comparing CSF-resistant and CSF-sensitive *C. auris* strains, where two CSF-resistant strains exhibited CFW hypersensitivity (21).

With respect to azole and AMB susceptibility, our overexpression and deletion data for Pdr16 indicate several cellular functions in fungal pathogens (Fig. S1E). In other fungi, Pdr16 operates as a putative phosphatidylinositol transfer protein acting at the interface of the endoplasmic reticulum (ER), Golgi, and mitochondrial membranes. Pdr16 may affect the distribution of membrane lipids by altering the formation of intracellular lipid droplets (22–24). Therefore, Pdr16 may play a role in the communication of lipid membranes between intracellular compartments in *C. auris*. This was further illustrated by a lower susceptibility of e*PDR16* to antimycin A, a mitochondrial electron transport complex III inhibitor, as well as tunicamycin (TUN) and cerulenin (CER), which inhibit N-linked glycosylation and fatty acid biosynthesis at the ER, respectively (Fig. 1B, AA, TUN, CER). This correlated with the observation that strain R is slightly more resistant to CER when compared to the *pdr16Δ* deletion strain (Fig. 1B, CER). Of note, impaired mitochondrial and ER functions are associated with the production of reactive oxygen species under stress conditions. Such oxidative stress can cause organelle and DNA damage, thus explaining the increased susceptibility to AMB and 5FC (16). Furthermore, Pdr16 may regulate sphingolipid biogenesis and associated lipid signaling pathways. Indeed, Pdr16 levels impact the action of Aureobasidin A (AbA) (25), since AbA resistance was observed in e*PDR16* strains (Fig. 1B, AbA). Thus, e*PDR16* may alter the generation of sphingolipids, which could bind and sequester AMB in lipid bilayers, thereby dampening the AMB effect on ergosterol (26). Interestingly, we also observed that e*PDR16* exhibits better fitness in the presence of SDS when compared to the wild type (Fig. 1C), indicating a modulation of membrane integrity, which is consistent with earlier reports (27). Furthermore, e*PDR16* causes hypersensitivity to heat stress and osmolarity changes imposed by KCl and NaCl (Fig. 1C), suggesting a putative role of Pdr16 in osmostress response. Moreover, Pdr16 may play a role in modulating membrane fluidity by affecting lipid distribution between and within lipid bilayers, thereby altering antifungal sensitivities (28, 29). Overall, beyond overexpression, it is reasonable to speculate that differences in basal or regulated *PDR16* expression levels in distinct clinical isolates may influence antifungal susceptibility. Although some effects appear moderate, they imply several potential mechanisms contributing to *PDR16*-mediated azole susceptibility in *C. auris*.

Notably, the MIC changes observed in the S strain overexpressing Pdr16 (e*PDR16*) are greater than those seen in the *pdr16Δ*. Several possible explanations exist. First, the expression level of Pdr16 in R may be lower than in the e*PDR16* strain. Moreover, and as previously reported, strain R enhances distinct drug resistance mechanisms, including the upregulation of Cdr1 and Mdr1 responsible for azole resistance (21, 30–32), and the upregulation of Nce103 and Rca1 contributing to amphotericin B resistance (16). These additional resistance pathways in strain R are likely to mask the impact of *PDR16* deletion

on drug susceptibility. Furthermore, the results also imply that *C. auris* engages diverse and partially overlapping mechanisms to mount drug resistance.

Taken together, our letter reports a hitherto unrecognized role of Pdr16 in modulating the antifungal susceptibility of *Candida auris* to different drugs, especially azoles. Deletion of *PDR16* resulted in a 4-fold decrease in the MIC for azoles, while ectopic overexpression of this protein reduced susceptibility to different classes of antifungal drugs, including azoles, amphotericin B, and 5-fluorocytosine (5FC). The data suggest that *PDR16* expression levels in clinical isolates could serve as a surrogate marker to predict antifungal resistance traits.

## ACKNOWLEDGMENTS

We express our gratitude to all lab members for their technical support and experimental advice. We also thank Neeraj Chauhan and Anuradha Chowdhary for providing clinical isolates. This research was funded by grants from the Austrian Science Fund (FWF) to KK (ChromFunVir; P-32582-B08 and BacFun P-34152). TPC received support through a Student Fellowship through the Austrian Academic Exchange (Ernst Mach Grant 2021–2023), the Bernd Rode Award 2024 from the ASEA-UNINET network, an ESCMID Research Grant 2023 from the European Society of Clinical Microbiology and Infectious Diseases, and the FWF-funded PhD training program (TissueHome – FWF-DOC32-B28).

## AUTHOR AFFILIATIONS

[1]Max Perutz Labs, Vienna Biocenter Campus (VBC), Dr.-Bohr-Gasse 9, Vienna, Austria
[2]Medical University of Vienna, Center for Medical Biochemistry, Dr.-Bohr-Gasse 9, Vienna, Austria
[3]Labdia - Labordiagnostik GmbH, CCRI – St. Anna Children's Cancer Research Institute, Vienna, Austria

## AUTHOR ORCIDs

Trinh Phan-Canh  http://orcid.org/0000-0002-6399-0959
Karl Kuchler  http://orcid.org/0000-0003-2719-5955

## FUNDING

| Funder | Grant(s) | Author(s) |
| --- | --- | --- |
| Austrian Science Fund | ChromFunVir P-32582-B08, BacFun P-34152 | Karl Kuchler |
| ASEAN-European Academic University Network | Ernst Mach Grant 2021-2023, Bernd Rode Award 2024 | Trinh Phan-Canh |
| European Society of Clinical Microbiology and Infectious Diseases | ESCMID Research Grant 2023 | Trinh Phan-Canh |
| Austrian Science Fund | TissueHome - FWF-DOC32-B28 | Trinh Phan-Canh |

## AUTHOR CONTRIBUTIONS

Trinh Phan-Canh, Conceptualization, Data curation, Formal analysis, Funding acquisition, Investigation, Project administration, Visualization, Writing – original draft, Writing – review and editing, Methodology, Validation | Tamires Bitencourt, Investigation | Karl Kuchler, Conceptualization, Funding acquisition, Methodology, Project administration, Resources, Supervision, Writing – original draft, Writing – review and editing

## ADDITIONAL FILES

The following material is available online.

## Supplemental Material

**Supplemental material (Spectrum02659-24-S0001.docx).** Fig. S1; Table S1.

## Open Peer Review

**PEER REVIEW HISTORY (review-history.pdf).** An accounting of the reviewer comments and feedback.

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
