## [Reviewer comments · Microbiology Spectrum]

Microbiology Spectrum

Gene dosage of *PDR16* modulates azole susceptibility in *Candida auris*

Trinh Phan-Canh, Tamires Bitencourt, and Karl Kuchler

Corresponding Author(s): Karl Kuchler, Medizinische Universitat Wien

Review Timeline:

Submission Date:	October 25, 2024
Editorial Decision:	November 30, 2024
Revision Received:	December 17, 2024
Editorial Decision:	December 20, 2024
Revision Received:	February 10, 2025
Accepted:	February 28, 2025

Editor: Robert Arkowitz

Reviewer(s): The reviewers have opted to remain anonymous.

Transaction Report:

DOI: <https://doi.org/10.1128/spectrum.02659-24>

Re: Spectrum02659-24 (Gene dosage of *PDR16* modulates azole susceptibility in *Candida auris*)

Dear Mr. Trinh Canh Phan:

Thank you for the privilege of reviewing your work. Below you will find my comments, instructions from the Spectrum editorial office, and the reviewer comments.

Both reviewers found this short report to be of interest and, in general, the results support the conclusions. Reviewer #1 makes a number of minor comments that should not be an issue to resolve. Reviewer #2 has raised several points, the most critical being that the *pdr16* deletion mutant should be complemented by *PDR16* to ensure that the defects observed are indeed due to the absence of this gene. This critical control is important for interpretation of the results. Other comments from reviewer #2 will require additional details and discussion, on the whole. Overall these changes should improve this work and make it more accessible to readers.

Revision Guidelines

Sincerely,

Robert Arkowitz
Editor
Microbiology Spectrum

Reviewer #1 (Comments for the Author):

This short work addressed the effect of PDR16 on drug resistance in *C. auris*. Authors showed that the deletion of this gene as well as its overexpression result in changes of drug susceptibility depending on the strain background. Inferred from spotting assays on other drugs acting at the level of the cell wall or lipid biosynthesis, authors suggest that PDR16 may change cell membrane composition as well as organelle cross-talk within the cell and show the vast effects of PDR16 in *C. auris*. In conclusion, this work is concise and well-constructed and results support the hypothesis formulated by authors. Some points to clarify.

- 1) Author state "Specifically, ePDR16 increased MICs for azoles by 4- to 16-fold, while deletion of PDR16 in strain R reduced MICs by about four-fold". This statement is not that true for fluconazole as showed in Fig 1 in which fold-increase and decrease are different from 4- to 16-fold. Please correct
- 2) Authors state "Indeed, Pdr16 levels impact the action of Aureobasidin A (AbA) (25), since AbA hypo-sensitive was observed in ePDR16 strains". Please correct the term "AbA hypo-sensitive". Should that be simply "AbA-resistant"?
- 3) Authors state "we also observed that ePDR16 caused slight tolerance to SDS". How do you define tolerance to SDS? Please comment.
- 4) Authors state "it is reasonable to speculate that gain-of- function mutations in Pdr16 may alter antifungal susceptibility". How GOF can be achieved in PDR16? Authors could consider that PDR16 expression can be also modulated in clinical isolates by specific mutations in transcription factors. Please comment and modify statement accordingly.

Reviewer #2 (Comments for the Author):

The authors of this letter investigate the impact of PDR16 on drug susceptibility in *Candida auris* by testing the tolerance of a PDR16 deficient strain and a PDR16 overexpressing strain to different antifungals. They found that alteration of PDR16 expression, especially overexpression of the gene, increases resistance to antifungals when compared to its parental sensitive strain. They also found that this ePDR16 construct had increased susceptibility to some chemical stressors. While PDR16 and its involvement in drug resistance has been described before in *Candida albicans*, this gene has not been investigated in this context in *C. auris* and thus provides novel understanding of potential drug resistance mechanisms in this organism. However, the manuscript would be strengthened by addressing the following points:

1. The authors should assess and discuss the observed effects of deletion of PDR16 in the drug-resistant background. For example, it appears that ePDR16 increases drug tolerance to every drug tested except caspofungin. Why isn't the inverse true of the pdr16 Δ mutant, or at least not to the extremes of the ePDR16 mutant? For example, overexpression of PDR16 does increase fluconazole, amphotericin B, and flucytosine MIC, but deletion of PDR16 does not decrease the MIC in the presence of these drugs. More diligent discussion on how this could be occurring is warranted.
2. The authors created a pdr16 Δ strain, however a complement strain should also be constructed to ensure the observed phenotypes are driven by the absence of PDR16.
3. The ePDR16 overexpression construct is not a gain-of-function mutant as the PDR16 gene is simply being overexpressed, and no changes to protein function or activation has occurred.
4. For Figure S1B, more careful presentation of the data is needed, along with statistics. It's unclear what each sample is being compared to (WT S?). The inclusion of voriconazole in this figure is also a bit confusing, as it is never discussed in the main text.
5. Could at least some of the sensitivities and/or tolerances to the drugs tested in Figure 1C be due to overall morphological/fitness defects of the ePDR16 strain? This is being asked in part due to the fact that ePDR16 is hypersensitive to Congo red and calcofluor white despite the MIC between that strain and its parent being similar (regardless of paradoxical growth). Inclusion of the pdr16 Δ strain in the experiments conducted in Figure 1B & C may help discern whether the ePDR16 cells are unhealthy or if PDR16 is playing a role in cell wall integrity, lipid metabolism, etc.
6. It would be interesting to do some type of checkerboard growth assay with the azoles/amphotericin B and tunicamycin/cerulenin/etc. with the ePDR16 strain to test if, for example, inhibiting the ability to make fatty acids and sphingolipids renders this overexpression strain susceptible to the antifungals again.
7. An important overlooked phenotype is that the overexpression of PDR16 makes the cells sensitive to higher temperatures (Figure 1C). This may indicate that PDR16 plays a role in membrane fluidity, which could explain a number of the membrane and cell wall stress phenotypes. This should be addressed as a possible mechanism.

Other comments

1. Line 49: should say "particularly" instead of "particular."
2. In figure legend of Supplemental Figure 1, is the housekeeping gene supposed to be ACT1 and not ATC1?
3. What are the basal levels of PDR16 expression in the WT R and WT S strains?
4. Describe what MMS is in Figure legend.

RESPONSE TO REVIEWER COMMENTS

Reviewer #1 (Comments for the Author):

This short work addressed the effect of PDR16 on drug resistance in *C. auris*. Authors showed that the deletion of this gene as well as its overexpression result in changes of drug susceptibility depending on the strain background. Inferred from spotting assays on other drugs acting at the level of the cell wall or lipid biosynthesis, authors suggest that PDR16 may change cell membrane composition as well as organelle cross-talk within the cell and show the vast effects of PDR16 in *C. auris*. In conclusion, this work is concise and well-constructed and results support the hypothesis formulated by authors. Some points to clarify.

General Response: We are delighted that reviewer #1 recognizes the relevance of our data, as they advance our understanding of *C. auris* drug susceptibility. We believe this letter will bring significant attention to a broader community regarding the impact of *PDR16* on complex drug resistance phenomena. Overall, we fully agree with the constructive feedback from reviewer #1, and we have thus revised the manuscript accordingly.

1) Author state "Specifically, ePDR16 increased MICs for azoles by 4- to 16-fold, while deletion of PDR16 in strain R reduced MICs by about four-fold". This statement is not that true for fluconazole as showed in Fig 1 in which fold-increase and decrease are different from 4- to 16-fold. Please correct

Response: Thank you - we have made the correction. **Line 42:** "up to 16-fold."

2) Authors state "Indeed, Pdr16 levels impact the action of Aureobasidin A (AbA) (25), since AbA hypo-sensitive was observed in ePDR16 strains". Please correct the term "AbA hypo-sensitive". Should that be simply "AbA-resistant"?

Response: Yes indeed, we agree that "AbA-resistant" is a more appropriate wording.
Changes in revised manuscript, line 75-76: "... since AbA resistance was observed in ePDR16"

3) Authors state "we also observed that ePDR16 caused slight tolerance to SDS". How do you define tolerance to SDS? Please comment.

Response: We agree and have thus rephrased this sentence to avoid misunderstandings.
Changes in revised manuscript, line 78-79: "ePDR16 exhibits better fitness growing in the presence of SDS when compared to the wild type."

4) Authors state "it is reasonable to speculate that gain-of- function mutations in Pdr16 may alter antifungal susceptibility". How GOF can be achieved in PDR16? Authors could consider that PDR16 expression can be also modulated in clinical isolates by specific mutations in transcription factors. Please comment and modify statement accordingly.

Response: Yes, this is yet another possibility, reflecting complex backgrounds that promote drug resistance phenomena. We agree and have revised the wording as follows.
Changes in revised manuscript, line 85: "... alterations of *PDR16* expression levels among different clinical isolates may influence antifungal susceptibility"

Reviewer #2 (Comments for the Author):

The authors of this letter investigate the impact of PDR16 on drug susceptibility in *Candida auris* by testing the tolerance of a PDR16 deficient strain and a PDR16 overexpressing strain to different antifungals. They found that alteration of PDR16 expression, especially overexpression of the gene, increases resistance to antifungals when compared to its parental sensitive strain. They also found that this ePDR16 construct had increased susceptibility to some chemical stressors. While *PDR16* and its involvement in drug resistance has been described before in *Candida albicans*, this gene has not been investigated in this context in *C. auris* and thus provides novel understanding of potential drug resistance mechanisms in this organism. However, the manuscript would be strengthened by addressing the following points:

Response: We are pleased to receive positive and insightful feedback from reviewer #2, who also recognizes the relevance of our work. Further, we appreciate valuable experimental suggestions. Indeed, we are particularly interested in the impact of sphingolipid biosynthesis and the cross-talk with Pdr16 in establishing drug resistance traits of *Candida auris*. We agree to the notion that this pathway could hold promising targets for overcoming *C. auris* antifungal resistance (Kalra, Tanwar, and Bari 2024). Indeed, we have previously identified *PDR16* as a possible candidate target using a proteomic analysis of a drug-resistant clinical isolate (Phan-Canh, Trinh et al. 2024), highlighting a link of Pdr16 with sphingolipid biosynthesis and cell membrane integrity. However, our aim with this short letter limited to around 500 words was to rapidly communicate these significant observations. Given the underlying complexity, the time required for a complete mechanistic dissection is not predictable and certainly requires extensive further research. Thus, we have addressed the relevant points in detail as outlined below and added additional data into the revised manuscript to strengthen our report.

1. The authors should assess and discuss the observed effects of deletion of PDR16 in the drug-resistant background. For example, it appears that ePDR16 increases drug tolerance to every drug tested except caspofungin. Why isn't the inverse true of the *pdr16*Δ mutant, or at least not to the extremes of the ePDR16 mutant? For example, overexpression of PDR16 does increase fluconazole, amphotericin B, and flucytosine MIC, but deletion of PDR16 does not decrease the MIC in the presence of these drugs. More diligent discussion on how this could be occurring is warranted.

Response: We were initially surprised to observe the significant changes in the overexpression strain, particularly the remarkable increases in MIC values for amphotericin B (AMB) and azoles. We therefore performed extensive phenotyping of ePDR16 strain and the sensitive WT control S. The *PDR16* mRNA levels (Fig. S1B) were markedly upregulated in ePDR16 by ~64-fold, which is even higher than in the clinical R strain (Table R1). Hence, we generated a deletion mutant in a resistant strain background with a 6.7-fold increase in Pdr16 levels. This observation allows for two possible explanations. First, *PDR16* expression levels most likely vary depending on different strain backgrounds. Second, complex and interdependent mechanisms promoting drug resistance may operate in different clinical strains.

The latter seems more plausible, as the parental resistant strain (R) harbors multiple parameters that can contribute to a compound pan-antifungal resistance phenotype. First, the residue change S639F in *FKS1* leads to caspofungin resistance (Jenull Sabrina et al. 2022). Second, Cdr1 and Mdr1 are highly upregulated explaining the pronounced azole resistance (Phan-Canh, Trinh et al. 2024; Jenull et al. 2021). Finally, our unpublished work shows that elevated Nce103 levels controlled by the key transcription factor Rca1, control AMB resistance (see preprint: <https://doi.org/10.1101/2024.04.12.589292>, Fig. R1).

All in all, these findings indicated that *Candida auris* clinical isolates employ multiple strategies to evade action of all commonly used clinical antifungals. We have thus expanded on this matter in the discussion of the revised manuscript.

Changes in revised manuscript, line 88-95: Notably, the MIC changes observed in the S strain overexpressing Pdr16 (*ePDR16*) are greater than those seen in the *pdr16Δ*. Several possible explanations exist. First, the expression level of Pdr16 in strain R may be lower than *ePDR16*. Moreover, and as previously reported, the strain R enhances distinct drug resistance mechanisms, including the upregulation of Cdr1 and Mdr1 responsible for azole resistance (21, 30–32), and the upregulation of Nce103 and Rca1 contributing to amphotericin B resistance (16). These additional resistance pathways in strain R may mask the relative impact of *PDR16* deletion on drug susceptibility. Further, the results also imply that *C. auris* relies upon diverse and partially overlapping mechanisms to mount drug resistance.

Figure R1. A snapshot of Fig. S1 from our previous report (Phan-Canh, Trinh et al. 2024)

Table R1. log2FC of *PDR16* from RNA sequencing and proteomic data analysis on two clinical strains that used in this manuscript

geneID	GeneName	RNA seq Resistant vs Sensitive strains		Proteomics Resistant vs Sensitive strains	
		log2FC_RvsS	adj_P_RvsS	log2FC_RvsS	adj_P_RvsS
B9J08_004982	PDR16	0.329674214	0.006098685	2.756442518	0.039964652

2. The authors created a *pdr16Δ* strain, however a complement strain should also be constructed to ensure the observed phenotypes are driven by the absence of *PDR16*.

Response: While we normally agree that complemented strains are good controls, we believe that in the present case it is not necessary. First, we verified the genomic context of all deletions by 5', 3' PCR, including the neighboring genes. Second, we used a loss-of-gene PCR control to validate the ablation of *PDR16*. Most importantly, we performed drug susceptibility phenotyping on at least three independent transformants for the deletion mutant (Fig. R2). Below, we add the MIC data from independent transformants to demonstrate data consistency and reliability in support of our conclusions and for the inspection by the reviewer. Constructing complemented strains for all mutants lacking *PDR16* and repeating all phenotyping experiments would not be feasible within the revision timeline. We believe that this approach is reasonable and acceptable.

Figure R2. MIC assay for voriconazole and ketoconazole with independent *pdr16Δ* transformants in the WT and *rca1Δ* strain backgrounds yield highly similar results

3. The ePDR16 overexpression construct is not a gain-of-function mutant as the PDR16 gene is simply being overexpressed, and no changes to protein function or activation has occurred.

Response: Yes, we agree and thank you for noticing. We rephrased the sentence as also suggested by reviewer #1.

Changes in revised manuscript, line 85: "... alteration in PDR16 expression among clinical isolates may influence antifungal susceptibility".

4. For Figure S1B, more careful presentation of the data is needed, along with statistics. It's unclear what each sample is being compared to (WT S?). The inclusion of voriconazole in this figure is also a bit confusing, as it is never discussed in the main text.

Response: We apologize for being sloppy here, which was owing to the need for brevity. Statistical analysis has now been added to the figure. We also addressed the response to VRC treatment in line 47. The goal was to determine whether voriconazole alters the mRNA levels of selected genes in *ePDR16* when compared to the WT-S strain. The results suggest that mRNA levels are not significantly affected, implying that *PDR16* levels may influence intracellular crosstalk or protein translation rather than mRNA expression.

5. Could at least some of the sensitivities and/or tolerances to the drugs tested in Figure 1C be due to overall morphological/fitness defects of the ePDR16 strain? This is being asked in part due to the fact that ePDR16 is hypersensitive to Congo red and calcofluor white despite the MIC between that strain and its parent being similar (regardless of paradoxical growth). Inclusion of the *pdr16Δ* strain in the experiments conducted in Figure 1B & C may help discern whether the ePDR16 cells are unhealthy or if PDR16 is playing a role in cell wall integrity, lipid metabolism, etc.

Response: We appreciate this thoughtful and insightful suggestion. Indeed, we tested the *pdr16Δ* strain and its parental strain for sensitivity to cell wall perturbations (Fig. R2). The results showed only a slight effect of SDS, Congo red (CR), and calcofluor white (CFW), where *pdr16Δ* showed better growth in CR and CFW conditions, whereas *pdr16Δ* showed reduced growth in the presence of SDS. Although these observations may explain why *ePDR16* is hypersensitive to CR and CFW but more resistant to SDS, the effects are minimal and perhaps not biologically significant. As discussed earlier, *C. auris* likely engages multiple mechanisms in a combinatorial manner to set drug resistance traits across different clinical isolates. This supports the notion that the effects of mutations in different clinical backgrounds can be highly variable and complex.

For example, we knocked out several genes associated with AMB resistance in *C. auris*. However, the susceptibility data show significant variability across different clinical strains, highlighting the complexity and heterogeneity of drug resistance phenotypes in this pathogen.

The new data are now provided as Fig. S1C and updated Fig. 1B, next text in line 69-71: This correlates with the observation that strain R is slightly more resistant to CER when compared to the *pdr16*Δ deletion strain (**Fig. 1B - CER**).

Figure R3. Agar plate sensitivity assays at different temperatures in the presence and absence of inhibitors of biological processes, including cell wall function (Calcofluor White - CFW, Congo Red - CR), Osmosensitivity (potassium chloride - KCl, sodium chloride – NaCl), cell wall integrity (sodium dodecyl sulfate - SDS).

6. It would be interesting to do some type of checkerboard growth assay with the azoles/amphotericin B and tunicamycin/cerulenin/etc. with the ePDR16 strain to test if, for example, inhibiting the ability to make fatty acids and sphingolipids renders this overexpression strain susceptible to the antifungals again.

Response: Yes, indeed this is a great suggestion, and we are planning to do this systematically using our large set of clinical isolates from all known clades and compare the data to the present deletion mutants. Of note, the length restrictions applicable to the “letter” format preclude addition of such data.

7. An important overlooked phenotype is that the overexpression of PDR16 makes the cells sensitive to higher temperatures (Figure 1C). This may indicate that PDR16 plays a role in membrane fluidity, which could explain a number of the membrane and cell wall stress phenotypes. This should be addressed as a possible mechanism.

Response: We appreciate this point, on which we fully agree. We believe that the long-standing notion about the impact of membrane fluidity in drug susceptibility remains valid and Pdr16 could in fact also play a role in the dynamic distribution of membrane phospholipids (van den Hazel et al. 1999; Griac 2007). Hence, we discussed this in the manuscript.

Changes in revised manuscript, line 80-84: *Furthermore, ePDR16 caused hypersensitivity to heat stress and osmolarity changes by KCl and NaCl (Fig. 1C), suggesting a potential role of Pdr16 in osmostress response. Moreover, Pdr16 could play a role in modulating membrane fluidity by affecting lipid distribution between and within lipid bilayers, thereby altering drug sensitivities (28, 29).*

Other comments

1. Line 49: should say "particularly" instead of "particular."

Response: We apologize – corrected (Line 49).

2. In figure legend of Supplemental Figure 1, is the housekeeping gene supposed to be ACT1 and not ATC1?

Response: Yes, indeed – the typo was corrected.

3. What are the basal levels of PDR16 expression in the WT R and WT S strains?

Response: We discussed this in point 1 of the response letter and in Table R1.

4. Describe what MMS is in Figure legend.

Response: Sorry for the omission – MMS refers to methyl methane sulfonate - corrected.

References

1. Griac, Peter. 2007. "Sec14 related proteins in yeast." *Lipid Transporters in Cell Biology* 1771 (6): 737–45. <https://doi.org/10.1016/j.bbalip.2007.02.008>.
2. Hazel, H. Bart van den, Harald Pichler, Maria Adelaide do Valle Matta, Erich Leitner, André Goffeau, and Günther Daum. 1999. "PDR16 and PDR17, two homologous genes of *Saccharomyces cerevisiae*, affect lipid biosynthesis and resistance to multiple drugs *." *Journal of Biological Chemistry* 274 (4): 1934–41. <https://doi.org/10.1074/jbc.274.4.1934>.
3. Jenull Sabrina, Shivarathri Raju, Tsymala Irina, Penninger Philipp, Trinh Phan-Canh, Nogueira Filomena, Chauhan Manju, et al. 2022. "Transcriptomics and phenotyping define genetic signatures associated with echinocandin resistance in *Candida auris*." *mBio* 0 (0): e00799-22. <https://doi.org/10.1128/mbio.00799-22>.
4. Jenull, Sabrina, Michael Tschermer, Nataliya Kashko, Raju Shivarathri, Anton Stoiber, Manju Chauhan, Andriy Petryshyn, Neeraj Chauhan, and Karl Kuchler. 2021. "Transcriptome signatures predict phenotypic variations of *Candida auris*." *Frontiers in Cellular and Infection Microbiology* 11:288. <https://doi.org/10.3389/fcimb.2021.662563>.
5. Kalra, Sapna, Sunita Tanwar, and Vinay Kumar Bari. 2024. "Insights into the role of sphingolipids in antifungal drug resistance." *Fungal Biology Reviews* 47 (March):100342. <https://doi.org/10.1016/j.fbr.2023.100342>.
6. Phan-Canh, Trinh, Philipp Penninger, Saskia Seiser, Narakorn Khunweeraphong, Doris Moser, Tamires Bitencourt, Hossein Arzani, Weiqiang Chen, Lisa-Maria Zenz, and Andrej Knarr. 2024. "Carbon dioxide controls fungal fitness and skin tropism of *Candida auris*." *bioRxiv*, 2024–04.

Re: Spectrum02659-24R1 (Gene dosage of *PDR16* modulates azole susceptibility in *Candida auris*)

Dear Karl:

Thank you for the privilege of reviewing your work. Below you will find my comments, instructions from the Spectrum editorial office, and the reviewer comments.

Thank you for addressing a majority of the issues raised by the reviewers. Complementation of a loss of function mutant, is however, essential to validate mutant phenotypes. While I appreciate the PCR verification and the isolation of three independent transformants, these approaches do not rule out the possibility that secondary mutations or effects on neighboring genes underly the mutant phenotype. For example, a pre-existing secondary mutation in a cell population could end up in multiple transformants or it is possible that the generated mutation may result in a selection for a secondary mutation. As indicated in the initial review, in particular the comments of reviewer #2 and the decision letter, complementation of the *pdr16Δ* strain, which is a single strain, is critical to validate the increased susceptibility to voriconazole and ketoconazole observed with this mutant.

Revision Guidelines

Sincerely,

Rob Arkowitz
Editor
Microbiology Spectrum

RESPONSE TO REVIEWER COMMENTS

Reviewer #1 (Comments for the Author):

This short work addressed the effect of PDR16 on drug resistance in *C. auris*. Authors showed that the deletion of this gene as well as its overexpression result in changes of drug susceptibility depending on the strain background. Inferred from spotting assays on other drugs acting at the level of the cell wall or lipid biosynthesis, authors suggest that PDR16 may change cell membrane composition as well as organelle cross-talk within the cell and show the vast effects of PDR16 in *C. auris*. In conclusion, this work is concise and well-constructed and results support the hypothesis formulated by authors. Some points to clarify.

General Response: We are delighted that reviewer #1 recognizes the relevance of our data, as they advance our understanding about *C. auris* drug susceptibility. We believe this letter will bring significant attention to a broader community regarding the impact of *PDR16* on the complexity of antifungal resistance phenomena. Overall, we appreciate the constructive feedback from reviewer #1, and we have thus revised the manuscript accordingly.

1) Author state "Specifically, ePDR16 increased MICs for azoles by 4- to 16-fold, while deletion of PDR16 in strain R reduced MICs by about four-fold". This statement is not that true for fluconazole as showed in Fig 1 in which fold-increase and decrease are different from 4- to 16-fold. Please correct

Response: Thank you - we have made the correction. **Line 42:** "up to 16-fold."

2) Authors state "Indeed, Pdr16 levels impact the action of Aureobasidin A (AbA) (25), since AbA hypo-sensitive was observed in ePDR16 strains". Please correct the term "AbA hypo-sensitive". Should that be simply "AbA-resistant"?

Response: We agree that "AbA-resistant" is a more appropriate wording.

Changes in revised manuscript, line 75-76: "... since AbA resistance was observed in ePDR16"

3) Authors state "we also observed that ePDR16 caused slight tolerance to SDS". How do you define tolerance to SDS? Please comment.

Response: We have rephrased this sentence to avoid misunderstandings as follows.

Changes in revised manuscript, line 78-79: "ePDR16 exhibits better fitness in the presence of SDS when compared to the wild type."

4) Authors state "it is reasonable to speculate that gain-of- function mutations in Pdr16 may alter antifungal susceptibility". How GOF can be achieved in PDR16? Authors could consider that PDR16 expression can be also modulated in clinical isolates by specific mutations in transcription factors. Please comment and modify statement accordingly.

Response: This is indeed yet another possibility, reflecting the complexity of mechanistic input promoting drug resistance phenomena. We have revised the wording as follows:

Changes in revised manuscript, line 85: "... differences in basal or regulated *PDR16* expression levels in distinct clinical isolates may influence antifungal susceptibility."

Reviewer #2 (Comments for the Author):

The authors of this letter investigate the impact of PDR16 on drug susceptibility in *Candida auris* by testing the tolerance of a PDR16 deficient strain and a PDR16 overexpressing strain to different antifungals. They found that alteration of PDR16 expression, especially overexpression of the gene, increases resistance to antifungals when compared to its parental sensitive strain. They also found that this ePDR16 construct had increased susceptibility to some chemical stressors. While PDR16 and its involvement in drug resistance has been described before in *Candida albicans*, this gene has not been investigated in this context in *C. auris* and thus provides novel understanding of potential drug resistance mechanisms in this organism. However, the manuscript would be strengthened by addressing the following points:

Response: We appreciate the positive and insightful feedback from reviewer #2, who also recognizes the novelty and relevance of our work. Further, we appreciate valuable experimental suggestions. Of note, we agree that the sphingolipid biosynthesis pathway could hold promising targets for overcoming *C. auris* antifungal resistance (Kalra, Tanwar, and Bari 2024). Indeed, we have previously identified *PDR16* as a possible target through proteomics of a drug-resistant clinical isolate (Phan-Canh, Trinh et al. 2024), highlighting a link between Pdr16 and sphingolipid metabolism as well as membrane integrity. Our aim was to rapidly communicate these significant observations to the community. Thus, given the underlying complexity, the time required for a complete mechanistic dissection is far out of the scope for a short letter limited to around 500 words. Thus, we have addressed the relevant points in detail as outlined below and added additional data into the revised manuscript to strengthen the short report.

1. The authors should assess and discuss the observed effects of deletion of PDR16 in the drug-resistant background. For example, it appears that ePDR16 increases drug tolerance to every drug tested except caspofungin. Why isn't the inverse true of the *pdr16*Δ mutant, or at least not to the extremes of the ePDR16 mutant? For example, overexpression of PDR16 does increase fluconazole, amphotericin B, and flucytosine MIC, but deletion of PDR16 does not decrease the MIC in the presence of these drugs. More diligent discussion on how this could be occurring is warranted.

Response: We were initially surprised to observe significant MIC changes in the overexpression strain, particularly the remarkable increases for amphotericin B (AMB) and azoles. We therefore performed extensive phenotyping of ePDR16 strain and the sensitive WT control S. The *PDR16* mRNA levels (Fig. S1B) were upregulated in ePDR16 by ~64-fold, which is even higher than in the clinical R strain (Table R1). Hence, we generated a deletion mutant in a resistant background, resulting in a 6.7-fold increase of Pdr16 levels. This observation allows for two possible explanations. First, *PDR16* expression levels most likely vary depending on strain backgrounds. Second, complex and interdependent mechanisms promoting drug resistance may operate in different clinical strains. The latter seems more plausible, as the parental resistant strain (R) harbors multiple parameters that can contribute to a compound pan-antifungal resistance phenotype. First, the change S639F in *FKS1* leads to caspofungin resistance (Jenull Sabrina et al. 2022). Second, Cdr1 and Mdr1 are highly upregulated explaining the pronounced azole resistance (Fig. R1) (Phan-Canh, Trinh et al. 2024; Jenull et al. 2021). Finally, our unpublished work shows that elevated Nce103 levels, controlled by the key transcription factor Rca1, control AMB resistance (see preprint: <https://doi.org/10.1101/2024.04.12.589292>). All in all, these findings strongly suggest that *C. auris* clinical isolates employ multiple combinatorial strategies to evade commonly used antifungal drugs. We have thus expanded on this matter in the discussion of the revised manuscript.

Changes in revised manuscript, line 88-95: Notably, the MIC changes observed in the S strain overexpressing Pdr16 (ePDR16) are greater than those seen in the *pdr16*Δ. Several possible explanations exist. First, the expression level of Pdr16 in R may be lower than in the ePDR16 strain. Moreover, and as previously reported, strain R enhances distinct drug resistance

mechanisms, including the upregulation of Cdr1 and Mdr1 responsible for azole resistance (21, 30–32), and the upregulation of Nce103 and Rca1 contributing to amphotericin B resistance (16). These additional resistance pathways in strain R are likely to mask the impact of *PDR16* deletion on drug susceptibility. Further, the results also imply that *C. auris* engages diverse and partially overlapping mechanisms to mount drug resistance.

Figure R1. A snapshot of Fig. S1 from our previous report (Phan-Canh, Trinh et al. 2024)

Table R1. log₂FC of *PDR16* from RNA sequencing and proteomic data analysis on two clinical strains that used in this manuscript

geneID	Gene	RNA seq of R vs S strains		Proteomics of R vs S strains	
		log ₂ FC_RvsS	adj_P_RvsS	log ₂ FC_RvsS	adj_P_RvsS
B9J08_004982	PDR16	0.329674214	0.006098685	2.756442518	0.039964652

2. The authors created a *pdr16*Δ strain, however a complement strain should also be constructed to ensure the observed phenotypes are driven by the absence of *PDR16*.

Response: As requested by reviewer 2 and the editor, we complemented the *pdr16*Δ mutant. The new MIC results for voriconazole (VRC) and ketoconazole (KTC) are now added into Supplementary Figure 1C.

Figure R2. MIC assay for voriconazole and ketoconazole for complemented mutant. Three independent transformants yield similar results.

3. The ePDR16 overexpression construct is not a gain-of-function mutant as the PDR16 gene is simply being overexpressed, and no changes to protein function or activation has occurred.

Response: Well, we do not know since we have not directly verified Pdr16 function. To avoid misunderstandings, we rephrased the relevant sentence as also suggested by reviewer #1.

Changes in revised manuscript, line 85: "... differences in basal or regulated PDR16 expression levels in distinct clinical isolates may influence antifungal susceptibility".

4. For Figure S1B, more careful presentation of the data is needed, along with statistics. It's unclear what each sample is being compared to (WT S?). The inclusion of voriconazole in this figure is also a bit confusing, as it is never discussed in the main text.

Response: We apologize for being sloppy here, which was owing to the need for brevity. Statistical analysis has now been indicated in the figure legend. We also addressed the response to VRC treatment in line 47. The goal was to determine whether VRC alters mRNA levels of selected genes in the ePDR16 strain when compared to the WT-S strain. The results suggest that mRNA levels are not significantly affected, implying that PDR16 levels may influence intracellular crosstalk or protein translation rather than mRNA expression.

5. Could at least some of the sensitivities and/or tolerances to the drugs tested in Figure 1C be due to overall morphological/fitness defects of the ePDR16 strain? This is being asked in part due to the fact that ePDR16 is hypersensitive to Congo red and calcofluor white despite the MIC between that strain and its parent being similar (regardless of paradoxical growth). Inclusion of the pdr16Δ strain in the experiments conducted in Figure 1B & C may help discern whether the ePDR16 cells are unhealthy or if PDR16 is playing a role in cell wall integrity, lipid metabolism, etc.

Response: We appreciate this thoughtful and insightful suggestion. Indeed, we tested the pdr16Δ strain and its parental strain for altered sensitivity to cell wall perturbations (Fig. R3). The results showed only a slight effect for SDS, Congo red (CR), and calcofluor white (CFW), where pdr16Δ showed better growth in CR and CFW conditions, whereas pdr16Δ showed reduced growth in the presence of SDS. Although these observations may explain why ePDR16 is hypersensitive to CR and CFW but more resistant to SDS, the effects are minimal and perhaps not biologically relevant. As discussed earlier, *C. auris* likely engages multiple mechanisms in a combinatorial manner to establish drug resistance traits in different clinical isolates. This supports the notion that the effects of mutations in different clinical backgrounds can be highly variable and complex. For example, we deleted several genes putatively associated with AMB resistance in *C. auris*. However, the susceptibility data show significant variations across different clinical strains, highlighting the complexity and heterogeneity of drug resistance phenotypes in this pathogen.

The new data are now provided as Fig. S1C and updated Fig. 1B, next text in line 69-71: This correlates with the observation that strain R is slightly more resistant to CER when compared to the pdr16Δ deletion strain (Fig. 1B - CER).

Figure R3. Agar plate sensitivity assays at different temperatures in the presence and absence of stressors, including this affecting cell wall function (Calcofluor White - CFW, Congo Red - CR), osmosensitivity (potassium chloride - KCl, sodium chloride – NaCl), as well as cell wall integrity (sodium dodecyl sulfate - SDS).

6. It would be interesting to do some type of checkerboard growth assay with the azoles/amphotericin B and tunicamycin/cerulenin/etc. with the ePDR16 strain to test if, for example, inhibiting the ability to make fatty acids and sphingolipids renders this overexpression strain susceptible to the antifungals again.

Response: Yes, indeed this is a great suggestion. In fact, we are planning to do this in a systematic manner using our large set of clinical isolates from all known clades, and compare the data to all available deletion mutants. Of note, the length restrictions applicable to the “letter” format preclude addition of such data.

7. An important overlooked phenotype is that the overexpression of PDR16 makes the cells sensitive to higher temperatures (Figure 1C). This may indicate that PDR16 plays a role in membrane fluidity, which could explain a number of the membrane and cell wall stress phenotypes. This should be addressed as a possible mechanism.

Response: We appreciate this valid point. We believe that the long-standing notion that membrane fluidity is a critical factor for setting drug susceptibilities remains valid. For example, Pdr16 could play a role in the dynamics of lateral and transversal distribution of membrane phospholipids (van den Hazel et al. 1999; Griac 2007). Hence, we discussed this in the manuscript.

Changes in revised manuscript, line 80-84: Furthermore, ePDR16 causes hypersensitivity to heat stress and osmolarity changes imposed by KCl and NaCl (Fig. 1C), suggesting a putative role of Pdr16 in osmostress response. Moreover, Pdr16 may play a role in modulating membrane fluidity by affecting lipid distribution between and within lipid bilayers, thereby altering antifungal sensitivities (28, 29).

Other comments

1. Line 49: should say "particularly" instead of "particular."

Response: We apologize – corrected (Line 49).

2. In figure legend of Supplemental Figure 1, is the housekeeping gene supposed to be ACT1 and not ATC1?

Response: Yes, indeed – the typo was corrected.

3. What are the basal levels of PDR16 expression in the WT R and WT S strains?

Response: We discussed this in point 1 of the response letter and in Table R1.

4. Describe what MMS is in Figure legend.

Response: Sorry for the omission – MMS refers to methyl methane sulfonate - corrected.

References

1. Griac, Peter. 2007. "Sec14 related proteins in yeast." *Lipid Transporters in Cell Biology* 1771 (6): 737–45. <https://doi.org/10.1016/j.bbalip.2007.02.008>.
2. Hazel, H. Bart van den, Harald Pichler, Maria Adelaide do Valle Matta, Erich Leitner, André Goffeau, and Günther Daum. 1999. "*PDR16* and *PDR17*, two homologous genes of *Saccharomyces cerevisiae*, affect lipid biosynthesis and resistance to multiple drugs *." *Journal of Biological Chemistry* 274 (4): 1934–41. <https://doi.org/10.1074/jbc.274.4.1934>.
3. Jenull Sabrina, Shivarathri Raju, Tsymala Irina, Penninger Philipp, Trinh Phan-Canh, Nogueira Filomena, Chauhan Manju, et al. 2022. "Transcriptomics and phenotyping define genetic signatures associated with echinocandin resistance in *Candida auris*." *mBio* 0 (0): e00799-22. <https://doi.org/10.1128/mbio.00799-22>.
4. Jenull, Sabrina, Michael Tschermer, Nataliya Kashko, Raju Shivarathri, Anton Stoiber, Manju Chauhan, Andriy Petryshyn, Neeraj Chauhan, and Karl Kuchler. 2021. "Transcriptome signatures predict phenotypic variations of *Candida auris*." *Frontiers in Cellular and Infection Microbiology* 11:288. <https://doi.org/10.3389/fcimb.2021.662563>.
5. Kalra, Sapna, Sunita Tanwar, and Vinay Kumar Bari. 2024. "Insights into the role of sphingolipids in antifungal drug resistance." *Fungal Biology Reviews* 47 (March):100342. <https://doi.org/10.1016/j.fbr.2023.100342>.
6. Phan-Canh, Trinh, Philipp Penninger, Saskia Seiser, Narakorn Khunweeraphong, Doris Moser, Tamires Bitencourt, Hossein Arzani, Weiqiang Chen, Lisa-Maria Zenz, and Andrej Knarr. 2024. "Carbon dioxide controls fungal fitness and skin tropism of *Candida auris*." *bioRxiv*, 2024–04.

Re: Spectrum02659-24R2 (Gene dosage of *PDR16* modulates azole susceptibility in *Candida auris*)

Dear Karl:

Thank you for addressing the reviewers comments. Your manuscript has been accepted, and I am forwarding it to the ASM production staff for publication. Your paper will first be checked to make sure all elements meet the technical requirements. ASM staff will contact you if anything needs to be revised before copyediting and production can begin. Otherwise, you will be notified when your proofs are ready to be viewed.

Thank you again for submitting your paper to Spectrum.

Sincerely,
Rob Arkowitz
Editor
Microbiology Spectrum